# Unraveling the molecular pathobiology of vocal fold systemic dehydration using an *in vivo* rabbit model

**Naila Cannes do Nascimento**[1]*, **Andrea P. dos Santos**[2], **M. Preeti Sivasankar**[1], **Abigail Cox**[2]*

**1** Department of Speech, Language, and Hearing Sciences, Purdue University, West Lafayette, Indiana, United States of America, **2** Department of Comparative Pathobiology, Purdue University, West Lafayette, Indiana, United States of America

* ncannes@purdue.edu (NCN); adcox@purdue.edu (AC)

**Data Availability Statement:** The raw RNA-seq data underlying this study were deposited in the NCBI Gene Expression Omnibus (https://www.

## Abstract

Vocal folds are a viscoelastic multilayered structure responsible for voice production. Vocal fold epithelial damage may weaken the protection of deeper layers of lamina propria and thyroarytenoid muscle and impair voice production. Systemic dehydration can adversely affect vocal function by creating suboptimal biomechanical conditions for vocal fold vibration. However, the molecular pathobiology of systemically dehydrated vocal folds is poorly understood. We used an *in vivo* rabbit model to investigate the complete gene expression profile of systemically dehydrated vocal folds. The RNA-Seq based transcriptome revealed 203 differentially expressed (DE) vocal fold genes due to systemic dehydration. Interestingly, function enrichment analysis showed downregulation of genes involved in cell adhesion, cell junction, inflammation, and upregulation of genes involved in cell proliferation. RT-qPCR validation was performed for a subset of DE genes and confirmed the downregulation of *DSG1*, *CDH3*, *NECTIN1*, *SDC1*, *S100A9*, *SPINK5*, *ECM1*, *IL1A*, and *IL36A* genes. In addition, the upregulation of the transcription factor *NR4A3* gene involved in epithelial cell proliferation was validated. Taken together, these results suggest an alteration of the vocal fold epithelial barrier independent of inflammation, which could indicate a disruption and remodeling of the epithelial barrier integrity. This transcriptome provides a first global picture of the molecular changes in vocal fold tissue in response to systemic dehydration. The alterations observed at the transcriptional level help to understand the pathobiology of dehydration in voice function and highlight the benefits of hydration in voice therapy.

## Introduction

Vocal folds are a viscoelastic multilayered structure located in the larynx composed of a stratified squamous epithelium and lamina propria overlying the thyroarytenoid muscle [1, 2]. The epithelium is the outermost layer of the vocal fold and, together with the mucus, is the first barrier to protect the vocal folds from insults [3, 4]. It serves as an essential defense mechanism,

ncbi.nlm.nih.gov/geo/) under accession number GSE132765. All other relevant data are within the paper and its Supporting Information files.

**Funding:** This study was funded by the National Institutes of Health/National Institute on Deafness and other Communication Disorders (https://www.nidcd.nih.gov/). Grant R01DC015545 was awarded to MPS and AC. MPS and AC were co-principal investigators. The funder did not play a role in study design, data collection and analysis, decision to publish, or preparation of the manuscript.

**Competing interests:** The authors have declared that no competing interests exist.

and its association with voice health has become more understood over the past two decades [5–8]. The epithelial physical barrier is sustained by cell junctions, comprising tight junctions, adherens junctions, desmosomes, hemidesmosomes, and gap junctions. Cell junctions maintain tissue integrity by holding adjacent cells together and by anchoring the basal cell layer to the extracellular matrix; in addition, they regulate the paracellular transport of water and solutes [9, 10]. The maintenance of the epithelial barrier depends on the cell junction formation, distribution, and stability. Common, everyday insults, including irritants such as pollutants and laryngopharyngeal reflux (LPR), phonotrauma, and surgical procedures may cause epithelial barrier dysfunction, weakened vocal fold defense, and impaired voice production [6, 7, 11–16].

Maintaining hydration levels remains a core recommendation by voice specialists to sustain optimal vocal fold health and to prevent voice disorders such as vocal fatigue [17–22]. Systemic dehydration is characterized by reduced fluid within the body and can have many causes as simple as decreased water consumption and increased physical activity to more severe body fluid losses as a consequence of vomiting and diarrhea [23–27]. Experimentally, systemic dehydration is commonly induced by water withholding, with or without food access, or use of diuretics, or a combination of both [28–33]. Furosemide is a diuretic with a relatively fast onset of action that has been used to induce dehydration in numerous studies with animal and human subjects [25, 30, 34–38]. Regarding hydration and vocal folds, studies completed *ex vivo* and using animal and human subjects have shown that surface (i.e., fluid coating the vocal fold surface) and systemic hydration status have impacts on vocal fold biomechanics and physiology [12, 21, 30, 32, 39]. Although these studies provide some evidence on the benefits of hydration on vocal health, the impact of dehydration on laryngeal biology is still not fully elucidated [40]. Thus, unraveling the molecular mechanisms of dehydration is highly desired to support and possibly improve and personalize the standard clinical recommendation of increased hydration of the vocal folds [21, 22].

There is a paucity of molecular studies on vocal fold biology using *in vivo* models, with a number of studies analyzing the expression of specific genes or proteins based on their predicted functions rather than looking at the whole expression profile [41–44]. Despite the increase of publications on vocal folds transcriptome (microarray or RNA sequencing-based) and proteome analysis, most of the published studies are focused on investigating the expression profiles of challenged vocal fold fibroblasts alone [45–50] or normal mucosa [51]. A comprehensive investigation of the pathobiology of vocal fold tissue in response to insults has yet to be explored.

The first step to unveil the underlying mechanisms of vocal fold response to a given challenge is to apply a high-throughput approach to identify global molecular signatures instead of investigating the effect of single genes or proteins. Second, the analysis of the entire tissue would help to understand the role of each vocal fold layer in the presence of such challenge. And last but not least, the use of an *in vivo* model would reflect the response of the tissue interacting with the system as a whole. These three steps combined would provide a physiologically realistic picture of the impact of a targeted condition in the vocal fold biology.

RNA sequencing (RNA-Seq) is a high-throughput technique based on deep-sequencing technology widely used to analyze gene expression nowadays [52]. In contrast to the hybridization-based microarray technique, RNA-Seq determines the sequence of each cDNA in a given sample. Moreover, RNA-Seq does not have the limitations of high background noise and lower dynamic range of detection due to background and saturation of signals seen on microarrays [52–54]. We hypothesized that systemic dehydration causes transcriptional changes in genes related to structure and biomechanical properties, responsible for optimal vocal fold function, in the different tissue layers of the vocal folds. Thus, our study aimed to apply the

RNA-Seq approach to evaluate the effects of systemic dehydration in vocal fold tissue at the transcriptional level. To obtain physiologically realistic and translatable results, we used an *in vivo* animal model. Rabbit vocal fold has been used as a translational model for phonation and wound healing studies [55–59]. Therefore, we used rabbits treated with furosemide to stimulate body fluid loss as an *in vivo* model of acute systemic dehydration. The complete transcriptome of vocal fold tissue from dehydrated rabbits was compared to vocal folds from control animals treated with saline. We present herein the differentially expressed gene profile of systemically dehydrated vocal folds, which is marked by the downregulation of cell adhesion components. This transcriptome dataset is a new resource to explore the vocal fold's molecular mechanisms in response to systemic dehydration and to understand the clinical recommendation of hydration therapy at a deeper level.

## Materials and methods

### Animals and study design

This study followed all recommendations in the Guide for the Care and Use of Laboratory Animals of the National Institutes of Health. The animal protocol was approved by the Purdue Animal Care and Use Committee under the number 1606001428.

Fourteen 6-month-old male New Zealand White rabbits, acquired from Covance Inc. (New Jersey, USA), were subjects in this study. Rabbits were kept in individual perforated solid-bottom cages, with enrichment toys, in optimal temperature (15–20°C) and humidity-controlled (~40%) room with a 12h light/12h dark schedule at Purdue University animal facility during the study. Individual housing is recommended for male rabbits, which should be separated from other males at sexual maturity (12 to 14 weeks) to avoid aggressive behavior, but maintaining visual and olfactory contact with other rabbits (https://www.nc3rs.org.uk/3rs-resources/housing-and-husbandry/rabbits) [60]. Socialization training, consisting of food enrichment (hay) and human interaction (petting) was part of the acclimatization protocol to reduce stress for the rabbits and facilitate animal handling at the time of the study [61]. Animals were fed Teklad Global Rabbit Diet 2031 (Envigo, Madison, WI, USA) and hay and received water *ad libitum* until the day of the experiment. After one week of acclimation, animals were randomly divided into control (N = 6) and dehydrated (N = 8) groups. To reduce variability in the baseline hydration levels, all animals were pre-hydrated with 0.1 M sucrose solution for 48h prior to the experiment. Sucrose water is preferred over regular drinking water in rabbits [62]. The average body weight of all rabbits on the day of the experiment was 3.44 Kg (range of 3.04 to 3.89 Kg). To induce systemic dehydration of 5% body weight fluid loss (accepted range of 4.5–5.5%), rabbits received an IP injection of 5.0 mg/Kg of furosemide (50 mg/mL) (Salix Pharmaceuticals, Inc., Bridgewater Township, NJ, USA) and food and water were withheld. Control rabbits received IP injection of saline, to prevent animal handling from being a confounding variable and an unwanted source of variation between groups, and had access to food and water *ad libitum*. The volume of injectates was on average 0.34 mL/rabbit per IP injection. Dehydrated rabbits were weighed hourly after the furosemide injection. Rabbits received a second injection if body weight loss was less than 4% after 4 hours based on a pilot study where we observed that animals decreased or ceased urination frequency after 4 hours from first furosemide administration. Euthanasia was completed when dehydrated rabbits achieved 4.5–5.5% body weight loss (after ~3 to 6 h); control rabbits were euthanized immediately after dehydrated rabbits.

### Blood collection and analyses

Whole blood samples were collected into heparinized tubes prior to IP injection and before euthanasia to evaluate systemic dehydration markers such as hematocrit (HCT), total plasma

proteins (TPP; g/dL), plus eleven blood analytes incorporated in the i-STAT Chem8+ cartridge (Abaxis by Zoetis Inc., Parsippany-Troy Hills, NJ, USA). The blood analytes evaluated included creatinine (mg/dL), blood urea nitrogen (BUN; mg/dL), glucose (mg/dL), sodium (mmol/L), chloride (mmol/L), potassium (mmol/L), total $CO_2$ (mmol/L), ionized calcium (iCa; mmol/L), anion gap (mmol/L), HCT (%), and hemoglobin (g/dL). Packed cell volume (PCV) was also measured using heparinized microhematocrit capillary tubes (Thermo Fisher Scientific, Waltham, MA, USA) centrifuged at $15,000 \times g$ for 2 minutes and immediately verified using a microhematocrit reader card. TPP was assessed on the plasma using a Reichert's VET 360 refractometer (Ametek Reichert Technologies, Depew, NY, USA). The blood analyte values were recorded pre and post-dehydration. Percentage change of the blood parameters was calculated as final value (post-dehydration)–initial (baseline)/ initial*100, and compared between groups to verify systemic dehydration.

## Vocal fold collection and RNA extraction

The larynges and proximal trachea of each rabbit were excised immediately following euthanasia (1.0 mL IV dose of Beuthanasia-D Special, Schering Plough Animal Health Corp. Union, NJ, USA) and placed in a Petri dish on ice. Forceps held each larynx at the level of the trachea. A sagittal cut through the cricoid cartilage along the posterior surface of the larynx exposed the vocal fold mucosa. The edges of the opened larynx were pinned onto a wax surface and placed under a dissection microscope. Arytenoid cartilages identified the transverse level of the glottis where the vocal folds reside in the larynx. The soft tissue of the vocal fold (mucosa and thyroarytenoid muscle) was excised bilaterally and in its full depth until reaching the thyroid cartilage on the anterior surface using microdissection scissors. Two sections of approximately 3 mm length X 2 mm depth were immediately placed into sterile tubes containing RNAlater® Stabilization Solution (Invitrogen™ by Thermo Fisher Scientific) and stored at -80°C until RNA extraction. The entire microdissection procedure was accomplished in about 5 minutes per larynx. Total RNA was extracted using RNeasy Fibrous Tissue Mini Kit, including on-column DNAse I digestion step (QIAGEN, Hilden, Germany), following manufacturer's instructions. The concentration and quality of RNA were assessed by spectrophotometry (NanoDrop™, Thermo Fisher Scientific) and Agilent 2100 Bioanalyzer with an Agilent RNA 6000 Nano Kit (Agilent Technologies, Inc., Santa Clara, CA, USA).

## RNA sequencing and differential gene expression analysis

Ten vocal fold samples (4 control and 6 dehydrated) were used for RNA sequencing (RNA--Seq). All RNA samples had an RNA Integrity Number (RIN) of 7.3 or higher (Agilent 2100 Bioanalyzer). Library construction (RNA polyA) and RNA-Seq were performed by the Purdue Genomics Core Facility using Illumina NovaSeq (Illumina Inc., San Diego, CA, USA). Briefly, the Universal Plus mRNA-Seq with NuQuant library preparation kit (NuGEN Technologies, Inc., Tecan Group Ltd., Männedorf, Switzerland) was used to construct the libraries (200 ng of RNA/library) as directed by the kit manual, except that the RNA fragmentation was performed for 4 minutes, instead of 8 minutes to favor the generation of somewhat larger cDNA fragments. RNA-Seq was generated on an Illumina NovaSeq™ 6000 Sequencing System (Illumina) using a S4 flow cell and 300 cycles paired-end (2x150) chemistry.

The Bioinformatics Core Facility at Purdue University performed the differential gene expression analysis. Sequence quality assessment and trimming was done using FastQC (v 0.11.7) (https://www.bioinformatics.babraham.ac.uk/projects/fastqc/) and FASTX-Toolkit (v 0.0.14) (http://hannonlab.cshl.edu/fastx_toolkit/), respectively. Bases with a Phred33 score of less than 30 were removed, and the resulting reads with at least 50 bases of length were retained. The

quality trimmed reads were mapped against the reference genome of *Oryctolagus cuniculus* breed Thorbecke inbred, OryCun2.0/ENSSEMBL 95 (GenBank assembly accession: GCA_000003625.1) using STAR (v 2.5.4b) [63]. STAR mapping (bam) files were used for the analysis of differential gene expression by the Cuffdiff from Cufflinks (v 2.2.1) suite of programs [64]. Cuffdiff uses bam files to calculate Fragments per kilobase of exon per million fragments mapped (FPKM) values, from which differential gene expression between the pairwise comparisons (dehydrated versus control vocal folds) can be established. Gene annotation was retrieved from BioMart databases using biomartr package in 'R' (v 3.5.1; http://www.r-project.org/).

## Gene functional annotation and protein interaction network

Functional annotation of the differential expressed (DE) genes identified in the dehydrated group (with $p \leq 0.05$) was determined using the bioinformatics tool DAVID (v6.8) [65, 66] based on Gene Ontology (GO) and the Kyoto Encyclopedia of Genes and Genomes (KEGG) pathway enriched terms (https://www.genome.jp/kegg/pathway.html). GO terms were obtained from GO FAT enrichment annotation, which filters out very broad GO terms based on measured specificity of each term.

STRING (Search Tool for the Retrieval of Interacting Genes/Proteins) v.11.0 database was used to determine protein-protein interactions using the DE genes as input in order to visualize the relationship between these gene products based on different evidence levels including text-mining (co-mentioned in PubMed), experiments, curated databases, co-expression, neighborhood, gene fusion, and co-occurrence (https://string-db.org/). The interaction score (0 to 1) given by STRING represents an approximate confidence of the association between two proteins being true based on all the available evidence levels, and not the specificity or strength of an interaction [67].

## RNA-Seq validation by RT-qPCR

Primer pairs for qPCR were designed using Primer-BLAST tool [68] and selected for at least one primer of each pair to span an exon-exon junction to avoid amplification of residual genomic DNA (Table 1). A two-step reverse transcriptase quantitative PCR (RT-qPCR) was performed with SuperScript™ IV VILO™ Master Mix (Invitrogen™) for cDNA synthesis using 500 ng of total RNA per sample in a final reaction of 20 μL, following the manufacturer's protocol. Then, cDNA reactions were 10-fold diluted with RNAse/DNA-free water and added as 10% of the final qPCR volume (2.5 μL in 25 μL). The qPCR was run in triplicate per sample in a 96-well PCR plate using Power SYBR® Green PCR Master Mix (Applied Biosystems™ by Thermo Fisher Scientific), with a final concentration of 100 nM of each primer per reaction. The thermal cycling parameters for AmpliTaq Gold® DNA Polymerase were: 95 ˚C for 10 min; 40 cycles of 95 ˚C for 15 sec, and 60 ˚C for 1 min; and melt curve stage of 95 ˚C for 15 sec, 60 ˚C for 1 min, 95 ˚C for 1 sec. Reactions were carried out in a QuantStudio™ 3 Real-Time PCR System (Applied Biosystems™). A No-RT control containing RNA but no reverse transcriptase enzyme, and negative control with water instead of template were run in triplicate for each pair of primers on the qPCR plates. The relative expression level of each target gene was calculated with the $2^{-\Delta\Delta CT}$ method [69] using hypoxanthine phosphoribosyl-transferase 1 (*HPRT1*) as normalizer. All qPCR reactions had a single peak on the melt curve, verifying the amplification of a unique PCR product.

## Statistical analysis for hematologic and RT-qPCR data

Statistical analysis was performed using one-tailed Welch's *t*-test for the RT-qPCR data and Mann-Whitney nonparametric test for hematologic analytes to compare control

Table 1. Quantitative PCR primers for RNA-Seq validation.

| Gene symbol | Gene name | Forward (5' - 3') | Reverse (5' - 3') | Amplicon length (base pairs) |
|---|---|---|---|---|
| HPRT1* | hypoxanthine phosphoribosyl-transferase 1 | GATGGTCAAGGTCGCAAGCC | TCCAACAAAGTCTGGCCTGT | 73 |
| DSG1 | desmoglein 1 | TCCTGCTGGCATCGGATTAC | ATAGTGGCCAAACCAGTGGG | 195 |
| CDH3 | cadherin 3 | TGACAACCAAGAGGGGCTTG | ATCCTCTACGTGGACCACCA | 131 |
| CLDN7 | claudin 7 | TACGACTCTGTGCTCGCCCT | CAGCAAGACCTGCCACGATGAA | 199 |
| NECTIN1 (PVRL1) | nectin cell adhesion molecule 1 | AGTACCACTGGACCACGCTG | AGGAGACGGGGTGTAGGGAA | 191 |
| CAMSAP3 | calmodulin regulated spectrin associated protein family member 3 | GCCCGAGTACACAGGTCCTC | CGTGTACAGGGCTCGGAACT | 211 |
| PLEKHA7 | pleckstrin homology domain containing A7 | CAATGAGGAGGCGGCTACGA | GGCTCCACCCACCAGAGTTT | 111 |
| ECM1 | extracellular matrix protein 1 | GGCAGCCATCCCCGAACAA | GGGAGCTGGCTCTTCTTCTGT | 214 |
| SDC1 | syndecan 1 | GGGAGCCGGACTTCACTTTC | GCTGCCTTCGTCCTTCTTCT | 236 |
| GJB2 | Gap junction protein beta 2 | TTGGGGTGCGTGAGTGATGT | CTGCGCTTGCCACCAGTAAC | 78 |
| S100A9 | S100 calcium binding protein A9 | CCTCAAGAAGGAGGCGAGGG | AGCTGCTTGTCCTGGTTCGT | 78 |
| SPINK5 | serine peptidase inhibitor, Kazal type 5 | TGTGGAGATGATGGCCAGACG | ATTTGTCAGATGCAGGCAGGC | 154 |
| NR4A3 (NOR1) | nuclear receptor subfamily 4 group A member 3 | TTCTGACGGCCTCCATTGAC | AGCAGTGTTCGACCTGATGG | 149 |
| IL1A | interleukin 1 alpha | ATCTGGGCGATGCAGTGAAA | CCTGGGTGTCTCAGGCATTT | 158 |
| IL36A | interleukin 36 alpha | CAGGTGTGGGTCGTTCAGGA | GCTAACAGTGGCTGGAACCAT | 75 |

*HPRT1 was used as the endogenous control to calculate the relative expression levels of each tested gene.

versus dehydrated group. Grubbs' test was used to identify and remove outliers of each RT-qPCR pairwise analysis. All tests were executed with GraphPad Prism (version 6.0e for Mac). Differences between groups were considered statistically significant when p-value $\leq 0.05$.

## Results

### *In vivo* systemic dehydration model

Rabbits reached an average of 4.8% dehydration, based on body weight fluid loss, within 3 to 6 hours after 1 or 2 injections of furosemide. We aimed for a 5% body weight dehydration to translate our findings to a level of mild-moderate dehydration reported in the literature [24, 33, 70]. Control rabbits had a range of body weight loss of 0.2–1.8% (mean = 0.9%) caused by regular urination and bowel movement, and possibly by reduced food and water consumption due to handling stress (Fig 1A; p = 0.0003). The % changes in PCV, hemoglobin, and TPP were significantly different between control and dehydrated rabbits, with post-dehydration values consistently higher on the dehydrated group (Fig 1B). PCV values obtained by centrifugation and HCT obtained from the iSTAT were similar and had the same p-value on Mann-Whitney comparison analyses (p = 0.0003). In addition, blood creatinine and BUN % changes were significantly higher in the dehydrated rabbits compared to controls (Fig 1C). Sodium and chloride % changes in the blood were also significantly different between groups; however, post-dehydration levels in the dehydrated group were lower (Fig 1D), consistent with the furosemide mechanism of action that inhibits $Na^+$ and $Cl^-$ reabsorption [37]. The remaining blood analytes (glucose, potassium, total $CO_2$, iCa, and anion gap) tested by iSTAT blood analyzer did not change between groups. A summary of the Mann-Whitney results of all blood analytes is shown in S1 Table.

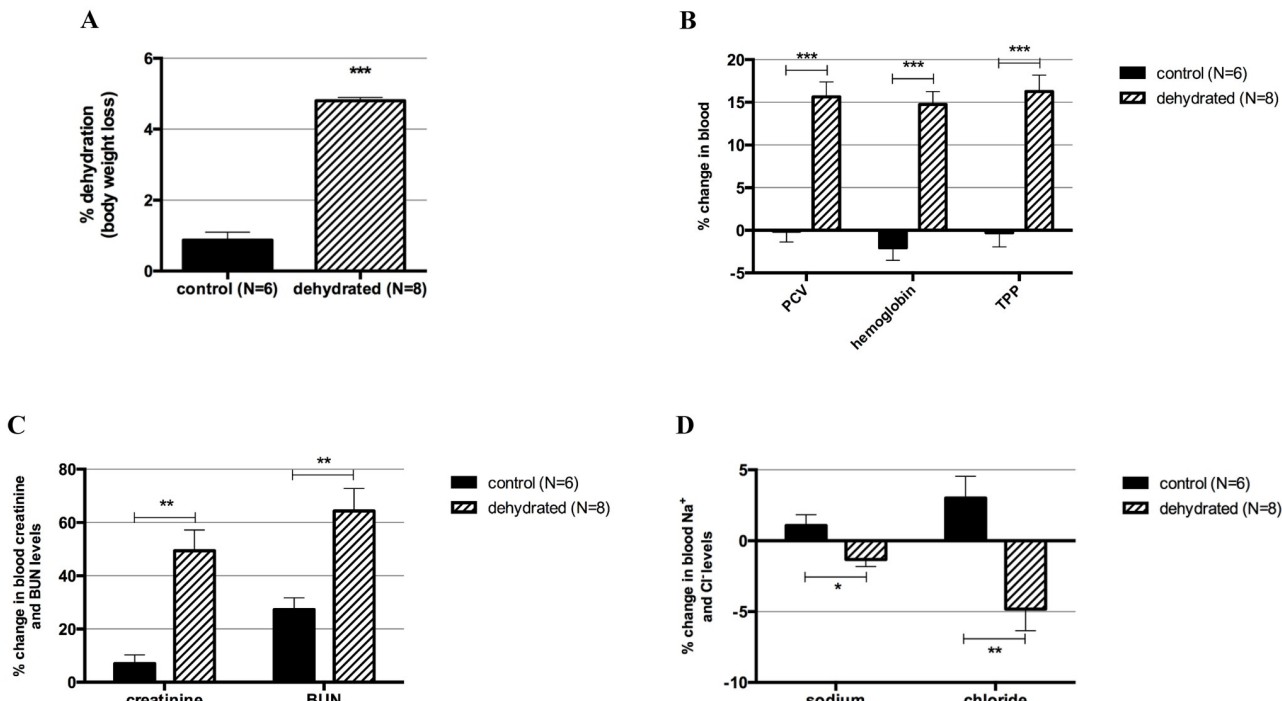

**Fig 1. Systemic dehydration markers.** Systemic dehydration was verified by body weight loss and blood markers. (A) Body weight loss in the control and dehydrated groups. (B) Percent changes in packed cell volume (PCV), hemoglobin, and total plasma proteins (TPP) in the control and dehydrated groups. (C) Percent changes in blood creatinine and blood urea nitrogen (BUN) levels in control and dehydrated groups. (D) Percent changes in blood sodium and chloride levels in control and dehydrated groups. Bars show mean ± SEM. ***p-value ≤ 0.001; **p-value ≤ 0.01; *p-value ≤ 0.05.

## Dehydrated vocal folds transcriptome

RNA-Seq data files were submitted to Gene Expression Omnibus (GEO, NCBI, https://www.ncbi.nlm.nih.gov/geo/) data repository under the accession number GSE132765. The average of total reads pair generated was 90,680,466 per sample, with more than 99% of read pairs passing quality control for all ten samples. A summary of the quality and mapping statistics of the reads of each vocal fold sample is provided in S2 Table.

A total of 23,669 genes were identified in both transcriptomes of control and systemic dehydrated rabbits, including genes expressed in muscle, lamina propria, and epithelium. Among these, 203 protein-coding genes were identified as differentially expressed (DE) in the vocal folds of dehydrated rabbits using the Cuffdiff method with p ≤ 0.05. Based on fold change, 152 DE genes were found downregulated in contrast to 51 upregulated. Only 12 out of these 203 DE genes have unknown products (S3 Table). DAVID functional annotation analysis recognized 185 out of 203 gene IDs; the remaining 18 were unmapped on the DAVID database. Functional classification of these genes based on GO terms FAT annotation and KEGG pathway enrichment analysis revealed 161 biological terms, divided into biological process (BP; 121 terms), cellular component (CC; 20 terms), molecular function (MF; 18 terms), and KEGG pathway (2 terms), with at least two DE genes per enriched term, and certain genes listed under multiple terms (Fig 2A and S4 Table). A number of the DE genes identified herein are related to epithelial processes, which directed the focus of the study to the vocal fold epithelium layer.

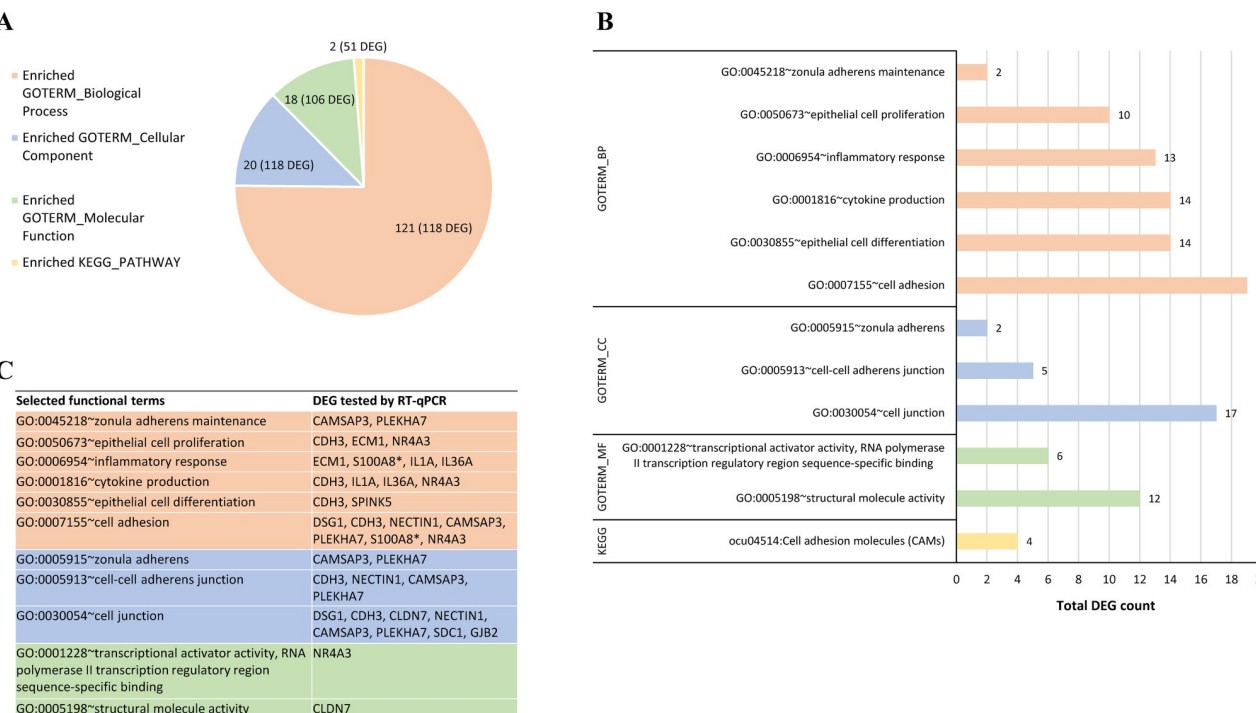

**Fig 2. Functional classification of differentially expressed genes (DEG) in dehydrated vocal folds by GO terms and KEGG pathway enrichment analysis.** (A) Number of enriched functional terms and total annotated DEG (in parenthesis) in each main category of biological process (BP), cellular component (CC), molecular function (MF), and KEGG pathway using DAVID analysis. (B) Enriched terms selected for further validation of DEG. Bars represent the total number of DEG per functional term. (C) DEG tested by RT-qPCR in each selected functional term. *S100A8 represents S100A9, which was tested by qPCR but not mapped by DAVID analysis; both share the same functions. The complete list of 161 enriched terms by functional category with associated DE genes is provided in S4 Table.

## RT-qPCR validation

Fourteen genes were selected for qPCR validation based on RNA-Seq fold change $\geq \pm 1.9$ and biological enrichment functions including cell adhesion (GO:0007155), epithelial cell differentiation (GO:0030855), inflammatory response (GO:0006954), epithelial cell proliferation (GO:0050673), zonula adherens maintenance (GO:0045218), cell junction (GO:0030054), cell adhesion molecules (CAMs) (ocu04514), among others (Fig 2B).

The DE genes tested comprise 13 downregulated genes in the dehydrated vocal fold group: eight cell junction-related genes, including desmoglein 1 (*DSG1*), cadherin 3 (*CDH3*), claudin 7 (*CLDN7*), nectin cell adhesion molecule 1 (*NECTIN1*), syndecan 1 (*SDC1*), calmodulin regulated spectrin associated protein family member 3 (*CAMSAP3*), pleckstrin homology domain containing A7 (*PLEKHA7*), and gap junction protein beta 2 (*GJB2*); two genes members of epidermal differentiation complex, S100 calcium-binding protein A9 (*S100A9*) and serine peptidase inhibitor, Kazal type 5 (*SPINK5*); two pro-inflammatory cytokines, interleukin 1 alpha (*IL1A*) and 36 alpha (*IL36A*), and the extracellular matrix protein 1 (*ECM1*) gene. In addition, the transcription factor nuclear receptor subfamily 4 group A member 3 (*NR4A3*) gene was chosen due to its association with the biological process of epithelial cell proliferation and its fold change of +23, indicating upregulation in the dehydrated group. The enriched functional terms and associated DE genes selected for RT-qPCR validation are shown in Fig 2C.

Differential expression of 10 out of 14 genes tested was validated by qPCR: *DSG1*, *CDH3*, *NECTIN1*, *SDC1*, *S100A9*, *SPINK5*, *ECM1*, *IL1A*, *IL36A*, and *NR4A3* (Fig 3). The remaining

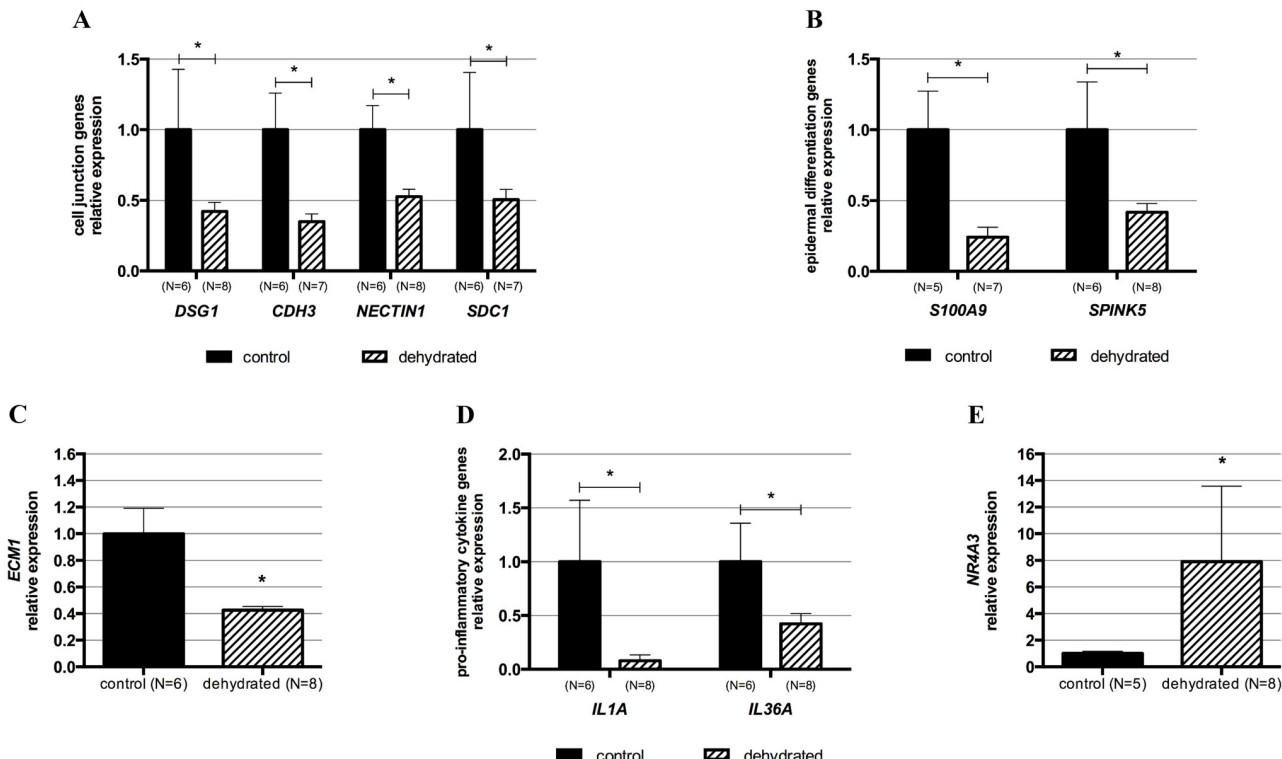

**Fig 3. RT-qPCR quantification of relative expression levels of selected differentially expressed genes in dehydrated vocal folds.** The differential expression of 10 genes in dehydrated vocal folds identified by RNA-Seq was validated by RT-qPCR. (A) Cell junction-related genes: desmoglein 1 (*DSG1*), cadherin 3 (*CDH3*), nectin cell adhesion molecule 1 (*NECTIN1*), and syndecan 1 (*SDC1*). (B) Epidermal differentiation-related genes: S100 calcium-binding protein A9 (*S100A9*) and serine peptidase inhibitor, Kazal type 5 (*SPINK5*). (C) Extracellular matrix protein 1 (*ECM1*) gene. (D) Pro-inflammatory cytokine genes: interleukin 1 alpha (*IL1A*) and 36 alpha (*IL36A*). (E) Transcription factor nuclear receptor subfamily 4 group A member 3 (*NR4A3*) gene. The gene expression levels of the control group were set to 1, and relative expression levels were calculated relative to the *HPRT1* gene using the method $2^{-\Delta\Delta Ct}$. Bars show mean ± SEM. *p-value $\leq$ 0.05. The detailed results of the qPCR analysis are listed in Table 2.

four genes identified as downregulated in the dehydrated group by RNA-Seq analysis were not statistically different from the control group by qPCR despite showing a trend of downregulation (S1 Fig). The qPCR data analysis is summarized in Table 2.

### Predicted protein interactions for the differentially expressed genes

STRING database was used to predict interactions between proteins encoded by the DE genes identified in the dehydrated group. Within the 203 DE genes identified, 117 showed predicted protein-protein interactions. Applying MCL clustering with a default inflation parameter of three [71], interactions were divided into 39 clusters with at least two proteins in each cluster, with direct associations (solid lines), indirect or inter-cluster associations (dashed lines). Twelve out of 14 proteins encoded by the DE genes tested by qPCR showed associations with each other or other proteins in the network (Fig 4). The products of *NECTIN1* and *NR4A3* did not show associations with other proteins in the network based on the evidence levels available on the STRING database.

## Discussion

Increasing systemic and superficial hydration are common clinical recommendations in the field of voice therapy to sustain healthy vocal folds and to prevent voice disorders [20, 22, 72,

**Table 2. Summary of qPCR results of selected differentially expressed genes in vocal folds of systemic dehydrated rabbits compared to controls.**

| Gene symbol | RNA-Seq fold change | qPCR fold change | qPCR p-value* |
| --- | --- | --- | --- |
| DSG1 | -1.99 | -2.37 | 0.0485 |
| CDH3 | -2.32 | -2.86 | 0.0224 |
| CLDN7 | -1.95 | -1.60 | 0.0704; ns |
| NECTIN1 | -2.23 | -1.89 | 0.0394 |
| CAMSAP3 | -1.97 | -1.66 | 0.0754; ns |
| PLEKHA7 | -1.99 | -2.23 | 0.0541; ns |
| ECM1 | -2.16 | -2.35 | 0.0191 |
| SDC1 | -2.19 | -1.97 | 0.0374 |
| GJB2 | -2.83 | -1.80 | 0.0854; ns |
| S100A9 | -6.63 | -4.14 | 0.0398 |
| SPINK5 | -2.75 | -2.39 | 0.0357 |
| NR4A3 (NOR1) | +23.76 | +7.92 | 0.0136 |
| IL1A | -3.65 | -12.72 | 0.0368 |
| IL36A | -2.92 | -2.37 | 0.0381 |

Negative and positive values of fold change indicate downregulation and upregulation of gene expression, respectively.

*Exact p-value of One-tailed Welch's *t*-test. ns: non-significant p-value.

73]. The beneficial effects of hydration and adverse consequences of dehydration on vocal fold physiology are fairly documented, particularly on phonatory threshold pressure, a measure of voice function [17, 18, 21, 30, 40, 73]. However, the pathobiology of systemic dehydration in this tissue is poorly understood. In our study, an *in vivo* systemic dehydration model using rabbits was developed to analyze the transcriptome of vocal folds in response to systemic dehydration. The rabbits received furosemide IP injection to induce systemic dehydration; furosemide alone or a combination with water withholding to study different effects of dehydration in rabbits are reported in the literature [29, 34]. Systemic dehydration was verified by body weight loss (4.8% average) and significant changes in the level of blood analytes compared to the control group. The higher values of PCV, hemoglobin, TPP, creatinine, and BUN post-dehydration are consistent with increased concentration of these analytes in the blood due to water loss in the urine stimulated by furosemide [74]. As expected, the post-dehydration levels of sodium and chloride in the blood of dehydrated rabbits decreased compared to baseline levels reflecting the mechanism of action of furosemide. As other loop diuretics, furosemide acts on the loop of Henle in the renal tubules inhibiting sodium and chloride reabsorption by binding to one of the $Cl^-$ binding sites of the $Na^+$-$K^+$-$2Cl^-$ cotransporter [37, 75]. In contrast, control rabbits, which received saline as a sham-injection, lost an average of 0.9% body weight and did not show changes in the levels of blood analytes pre and post-injection. These results together validate our model of acute systemic dehydration.

We used this *in vivo* rabbit model to investigate the transcriptional changes in vocal folds due to systemic dehydration. The RNA-Seq based transcriptome of dehydrated vocal folds revealed 152 downregulated and 51 upregulated genes. The functional classification of these DE genes was further explored to better understand the molecular impact of dehydration. Enrichment analysis using GO and KEGG annotations revealed 161 functional terms. We then focused on biological functions previously related to changes observed in vocal folds in response to other insults. These functions include cell adhesion, cell junction, epithelial cell proliferation, and inflammatory response, and are discussed below.

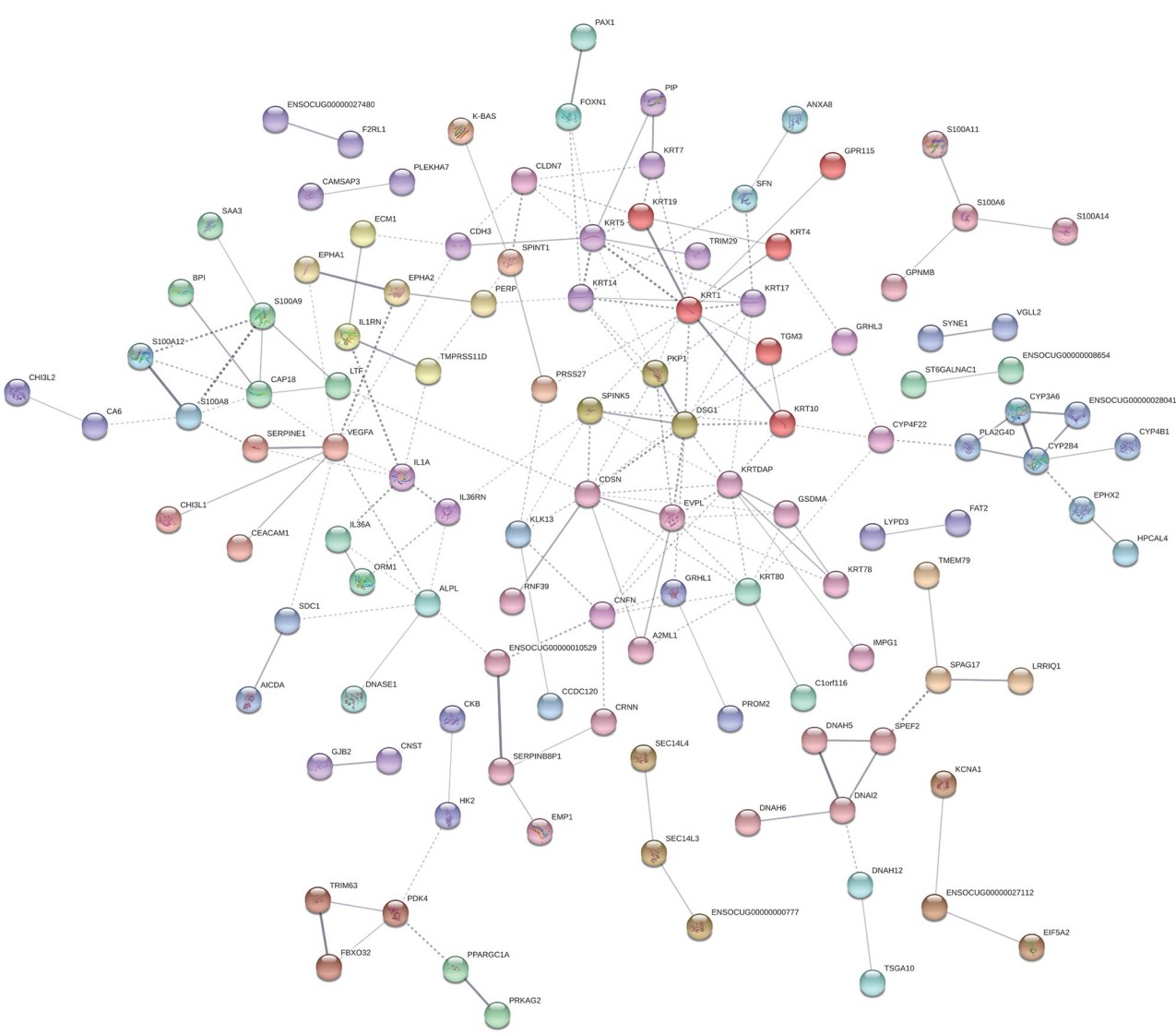

**Fig 4. STRING protein-protein interaction clusters for the differentially expressed (DE) genes identified in the vocal folds of systemically dehydrated rabbits.** A total of 117 differentially expressed (DE) genes had predicted protein-protein interactions. Each node represents a protein encoded by a DE gene. Network settings were: meaning of network clusters based on confidence (how many association evidences between proteins); minimum interaction score of 0.4 (medium confidence). Nodes with no predicted interactions were omitted from the network.

Our results suggest a perturbation in the vocal fold epithelium after an induced mild level of systemic dehydration [33] evidenced by the downregulation of 15 genes related to cell adhesion as well as 16 genes associated with cell junction in the transcriptome of the dehydrated group. Eight of these genes are involved in both cell adhesion and junction functions, and four were validated by qPCR, including *DSG1*, *CDH3*, *NECTIN1*, and *SDC1*. *DSG1* gene encodes the desmosomal cadherin desmoglein 1, an intercellular adhesion molecule localized primarily within the suprabasal epithelial layers. Desmoglein 1 is involved in maintaining epithelial homeostasis by regulating cell adhesion and supporting epithelial cell differentiation [76]. Cadherins and nectins, such as cadherin 3 (CDH3) and nectin 1 (NECTIN1), are components of epithelial adherens junctions, which are dynamic structures that undergo low and large-scale remodeling by substitution of adhesion molecules or disruption and reformation of

intercellular junctions [77]. Whether the downregulation of *CDH3* and *NECTIN1* genes in dehydrated vocal folds represent a remodeling without disrupting the intercellular adhesions or a junctional rearrangement impacting the epithelial integrity needs further investigation. Adherens junction and tight junction molecules are essential for the maintenance of the laryngeal epithelium structure, contributing to the protection and functional activity of this tissue [78]. Moreover, desmosomes are crucial for the integrity of tissues that undergo constant mechanical stress as do vocal folds [79, 80]. Interestingly, the literature reports alterations in the laryngeal epithelial barrier due to phonotrauma and LPR. A study with rabbits showed downregulation of gene expression of the tight junction occludin and the adherens junction β-catenin in the vocal folds after 30 minutes exposure to raised intensity phonation [57]. Besides, induced excessive phonation caused epithelium hyperplasia and surface epithelial cells shedding in feline vocal folds [81], while surface damage, including the destruction and loss of epithelial microvilli and desquamation, and marked tearing of desmosomes and hemidesmosomes were observed in canine vocal folds [82]. LPR was associated with reduced expression of E-cadherin and carbonic anhydrase isoenzyme III in the laryngeal epithelium of specimens from human patients, and exposure to acid solution and pepsin caused mucosal damage in an *in vitro* model of porcine larynx [5, 6]. These studies on phonotrauma and LPR, except for the human patients with LPR, share the acute character of our systemic dehydration model. Our results suggest that systemic dehydration, like phonotrauma and LPR, may be detrimental to the vocal fold epithelial barrier, which affects the structure of the tissue and, consequently, its normal function. Furthermore, systemic dehydration may prime or exacerbate the epithelial changes occurring during phonotrauma and LPR.

The RNA-Seq analysis also showed downregulation of 11 keratin genes. Keratins are markers of basal and suprabasal layers of the vocal fold epithelium [83, 84]. Epithelial cell layers undergo nearly constant turnover characterized by the continuous renewal of the basal and suprabasal cell layers [1, 85]. The reduced expression of keratin genes in the vocal folds observed in our animal model suggests that systemic dehydration alters this refined tissue equilibrium, likely impacting its normal structure and stability. In addition, the downregulation of *ECM1* (extracellular matrix protein 1) indicates that dehydration also interferes with the maintenance of the extracellular matrix. Interestingly, our group showed decreased hyaluronan amounts in the lamina propria of vocal folds of rats systemically dehydrated by water withholding [32]. Although the method of inducing systemic dehydration was different in that study, it shows the effect of this challenge in an extracellular matrix component that plays a key role in hydration homeostasis and viscoelastic properties of the vocal folds. It is important to note that the DE genes reported herein reflect the expression profile of all cell types in the laryngeal tissue analyzed. However, the most significant impact of dehydration on the epithelium is evidenced by differential expression of 20 genes related to cell adhesion and cell junctions, and 11 keratins.

The upregulation of the *NR4A3* gene was validated by qPCR. According to the GO function database, the protein encoded by this gene is involved in cell proliferation. NR4A3 and NR4A1 are homologous orphan nuclear receptors that regulate the expression of shared target genes. These transcription factors are involved in maintaining cellular homeostasis by regulating cell proliferation, differentiation, and apoptosis [86, 87]. The only study showing a regulatory role of an orphan nuclear receptor in vocal folds implicates NR4A1 as an endogenous inhibitor of fibrosis in rat vocal folds and human vocal fold fibroblasts [88]. In our model, the upregulation of *NR4A3* is likely associated with the activation of epithelial cell proliferation in an acute response scenario where fibrosis is not involved. Additional studies are warranted to understand the role of NR4A3 in the vocal fold epithelium as it relates to systemic dehydration.

We also observed the downregulation of genes that encode for members of the epidermal differentiation complex, including *S100A6*, *S100A8*, *S100A9*, *S100A11*, *S100A14*, in addition to *SPINK15* in the transcriptome of dehydrated vocal folds. Downregulation of *S100A9* and *SPINK5* were validated by qPCR. These genes contribute to the epithelial barrier maintenance by regulating epithelial growth and differentiation. In low concentrations, S100A8 and S100A9 might cause either tissue proliferation and repair, while in high concentrations, these proteins may have deleterious effects on inflamed tissue [89]. Moreover, S100A8 and S100A9 proteins act as nonchemokine chemoattractants of inflammatory cells [89, 90]. SPINK5 has been suggested to be an inhibitor of desquamation [91]; consequently, its downregulation could lead to increase cell turnover and cause epithelial barrier dysfunction [92]. Richer and collaborators (2009) showed a decreased expression of *S100A7*, *S100A8*, *S100A9*, and *SPINK5* genes in the airway mucosal epithelium of people with chronic rhinosinusitis, which they associated with a defective epithelial barrier in this condition [92]. Although this was observed in a chronic condition, it illustrates the association of the downregulation of these genes with epithelial barrier impairment. Finally, the blockage of S100A8 and S100A9 or downstream signaling was related to decreased pro-inflammatory cytokine secretion in several models [89]. In this context, the downregulation of these *S100* genes in our model could contribute in part to the downregulation of inflammatory-related genes, including the pro-inflammatory cytokines *IL1A* and *IL36A*. One may speculate that the reduced inflammatory response in vocal folds in response to systemic dehydration could help prevent tissue damage. Another hypothesis is that systemic dehydration could decrease the likelihood of vocal folds responding appropriately to an insult that requires inflammation such as wound healing or infection. Such hypotheses need to be further evaluated.

Our STRING protein network analysis showed that more than 100 of the proteins encoded by the DE genes identified by RNA-Seq interact with each other. Among these, DE genes discussed above including *DSG1*, *CDH3*, *SDC1*, *S100A9*, *SPINK5*, *ECM1*, *IL1A* and *IL36A*, are shown to interact with each other in the same cluster (same color code nodes in Fig 4) or indirectly with other clusters, suggesting a coordinated response of vocal fold to systemic dehydration. Interestingly, desmoglein 1 is one of the nodes with the highest number of interactions (11 total) with other products, suggesting that this cluster formed by *DSG1*, plakophilin 1 (*PKP1*), and *SPINK5* may play a central role in response to systemic dehydration. The importance of DSG1 for epithelial integrity is demonstrated by its role in several cutaneous diseases of different origins [76]. PKP1 interacts with desmosomal proteins and regulates desmosomal turnover and signaling [93], and the interaction between DSG1 and SPINK5 is evidenced by increased DSG1 degradation in mice deficient in *SPINK5* as a model of Netherton syndrome, a skin disorder [94]. This cluster interacts with two clusters of keratins, including KRT1, KRT10, KRT14, among others, which also show numerous interactions with other proteins in the network. The largest keratin cluster (red nodes) contains transglutaminase 3 (TGM3), which is expressed in squamous epithelia and involved in keratinocyte differentiation [95], and another downregulated gene in our model. Other products interacting with DSG1-PKP1-SPINK5 cluster are corneodesmosin (CDSN) and envoplakin (EVPL), both encoded by genes found downregulated in the systemic dehydrated vocal folds. CDSN and EVLP are both desmosomal components localized to epidermis and other cornified squamous epithelia. The loss of expression of both genes is associated with epidermal barrier defect leading to skin desquamation in human and murine model [96, 97]. Together, these interactions predicted between the DE genes along with their functional classification support the adverse impact of systemic dehydration, even at a low level of 5%, on the vocal fold epithelial structure. Thus, maintaining optimal systemic hydration may have a role in preserving the vocal fold tissue architecture and,

consequently, its normal function. How these transcriptional changes reflect in the proteome of vocal folds is our next focus of investigation.

## Conclusions

To our knowledge, this is the first study to analyze the global gene expression profile of vocal folds using an *in vivo* model of systemic dehydration. In our model, systemic dehydration altered the transcriptome of vocal folds by downregulating the gene expression of cell junction-related molecules, regulators of epithelial proliferation and differentiation, and keratins. These results suggest that systemic dehydration affects the epithelial homeostasis, and possibly causes dysregulation of the epithelial cell barrier. It is noteworthy that all the changes observed in our model were identified at a low level of systemic dehydration, highlighting the benefit of maintaining an optimal hydration status. Our transcriptome dataset provides a resource for the investigation of new hypotheses applying different approaches to continue elucidating the pathobiological effects of dehydration in vocal folds. Additional studies addressing the impact of systemic dehydration associated with other conditions detrimental to vocal function and health (e.g., phonotrauma and LPR) are warranted and may impact voice therapy practices in the future.

## Supporting information

**S1 Fig. RT-qPCR quantification of relative expression levels of selected genes in dehydrated vocal folds.** Genes: claudin 7 (*CLDN7*), calmodulin regulated spectrin associated protein family member 3 (*CAMSAP3*), pleckstrin homology domain containing A7 (*PLEKHA7*), and gap junction protein beta 2 (*GJB2*). The gene expression levels of the control group were set to 1, and relative expression levels were calculated relative to the *HPRT1* gene using the method $2^{-\Delta\Delta Ct}$. Bars show mean ± SEM. ns: non-significant.
(TIFF)

**S1 Table. Summary of pairwise analysis of hematologic analytes as markers of systemic dehydration.**
(PDF)

**S2 Table. Statistics of quality and mapping of the reads generated by RNA-Seq of vocal fold samples.** Columns description: **Sample ID**: sample names in the study; **Total Read Pairs**: number of total read pairs; **Quality Control Read Pairs**: number of read pairs after quality control, and % of reads that passed quality control; **Total Mapped Read Pairs**: number of reads mapped to the genome, and % of reads mapped to the genome; **Uniquely Mapped Read Pairs**: number of reads uniquely mapped to the genome, and % of reads uniquely mapped to the genome; **Read pairs went into genes**: number of reads in genic regions, and % of reads mapped to the genes.
(PDF)

**S3 Table. Differential expression analysis between control and dehydrated vocal fold samples using Cuffdiff (Cufflinks tool).** Only differentially expressed genes (q-value ≤ 0.05) are shown in the table. Columns description: **Ensembl gene ID**: describes the Ensembl gene identity; **Sample_1**: control; **Sample_2**: dehydrated; **Value_1**: FPKM of the gene in the control group; **Value_2**: FPKM of the gene in the dehydrated group; **log2fold_change**: describes log to base 2 value of fold change (Value_2/Value_1); **q-value**: The FDR-adjusted p-value of the test statistic; **Gene symbol**: official gene symbol; **Description**: annotation from Biomart. Note: All the log2 fold change values describe the fold change in the dehydrated group compared to control. Therefore, negative values represent downregulation, and positive values represent

the upregulation of a given gene in the dehydrated group.
(PDF)

**S4 Table. Functional categories of differentially expressed (DE) genes in dehydrated vocal folds by GO_TERM FAT and KEGG pathway enrichment analysis.** Columns description: Category: original database/resource of annotated terms (GOTERM_BP_FAT: biological process, GOTERM_CC_FAT: cellular component, GOTERM_MF_FAT: molecular function, KEGG_PATHWAY: KEGG pathway database); Term: enriched terms associated with gene list; Count: number of DE genes involved in the term; %: percentage of involved DE genes/ total DEG genes mapped on DAVID analysis, e.g., 28*100/185 = 15.13%; p-value: modified Fisher exact p-value, the smaller, the more enriched; DE genes list: Ensembl gene IDs of genes involved in the term; List total: number of genes in the gene list mapped to any term in this ontology; Pop hits: number of genes with this term on the background list (genes in the genome); Pop total: number of genes on the background list mapped to any term in this ontology; Fold enrichment: defined as the ratio of the two proportions (% DEG involved in a term/ % of background genes involved in the same term); FDR: False Discovery Rate.
(PDF)

## Acknowledgments

We are grateful to Jessica Engen, Taylor Bailey, and Chenwei Duan for their precious help with animal handling and samples collection. The authors also acknowledge the support of the Purdue Genomics Core Facility, especially Allison Sorg and Dr. Phillip SanMiguel, and the Bioinformatics Core at Purdue University, particularly Dr. Shaojun Xie and Dr. Jyothi Thimmapuram, for their services and support on data analysis.

## Author Contributions

**Conceptualization:** Naila Cannes do Nascimento, Andrea P. dos Santos, M. Preeti Sivasankar, Abigail Cox.

**Data curation:** Naila Cannes do Nascimento.

**Formal analysis:** Naila Cannes do Nascimento.

**Funding acquisition:** M. Preeti Sivasankar, Abigail Cox.

**Investigation:** Naila Cannes do Nascimento.

**Methodology:** Naila Cannes do Nascimento, Andrea P. dos Santos, M. Preeti Sivasankar, Abigail Cox.

**Project administration:** M. Preeti Sivasankar, Abigail Cox.

**Resources:** Andrea P. dos Santos, M. Preeti Sivasankar, Abigail Cox.

**Supervision:** M. Preeti Sivasankar, Abigail Cox.

**Validation:** Naila Cannes do Nascimento.

**Visualization:** Naila Cannes do Nascimento, Abigail Cox.

**Writing – original draft:** Naila Cannes do Nascimento.

**Writing – review & editing:** Naila Cannes do Nascimento, Andrea P. dos Santos, M. Preeti Sivasankar, Abigail Cox.

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
