## [Decision Letter · Decision Letter 0]

3 Jun 2020

PONE-D-20-11397

Unraveling the molecular pathobiology of vocal fold systemic dehydration using an in vivo rabbit model

PLOS ONE

Dear Dr. Cannes do Nascimento,

Thank you for submitting your manuscript to PLOS ONE. After careful consideration, we feel that it has merit but does not fully meet PLOS ONE’s publication criteria as it currently stands. Therefore, we invite you to submit a revised version of the manuscript that addresses the points raised during the review process.

We look forward to receiving your revised manuscript.

Kind regards,

Marie Jetté

Academic Editor

PLOS ONE

Journal Requirements:

2. In your Methods, please provide scientific justification for the individual housing of the rabbit.

3. We note that you are reporting an analysis of a microarray, next-generation sequencing, or deep sequencing data set. PLOS requires that authors comply with field-specific standards for preparation, recording, and deposition of data in repositories appropriate to their field. Please upload these data to a stable, public repository (such as ArrayExpress, Gene Expression Omnibus (GEO), DNA Data Bank of Japan (DDBJ), NCBI GenBank, NCBI Sequence Read Archive, or EMBL Nucleotide Sequence Database (ENA)). In your revised cover letter, please provide the relevant accession numbers that may be used to access these data. For a full list of recommended repositories, see http://journals.plos.org/plosone/s/data-availability#loc-omics or http://journals.plos.org/plosone/s/data-availability#loc-sequencing

Reviewers' comments:

Reviewer's Responses to Questions

**Comments to the Author**

1. Is the manuscript technically sound, and do the data support the conclusions?

Reviewer #1: Yes

Reviewer #2: Yes

Reviewer #3: Partly

2. Has the statistical analysis been performed appropriately and rigorously? 

Reviewer #1: Yes

Reviewer #2: Yes

Reviewer #3: Yes

3. Have the authors made all data underlying the findings in their manuscript fully available?

Reviewer #1: Yes

Reviewer #2: Yes

Reviewer #3: Yes

4. Is the manuscript presented in an intelligible fashion and written in standard English?

Reviewer #1: Yes

Reviewer #2: Yes

Reviewer #3: Yes

5. Review Comments to the Author

Reviewer #1: The main claims of this manuscript are that rapidly induced mild dehydration of rabbits induce molecular changes in vocal fold epithelium, and these changes are consistent with decreased homeostasis. Thus, the study seeks to elucidate a mechanism by which hydration can impact vocal fold integrity and, consequently, function. This work is significant in that it takes an incremental step towards identifying the molecular mechanism underlying well-established deleterious impacts of dehydration on vocal health (e.g. as stated on line 72). While vocal fold health depends on integrity of the epithelium, and underlying layers, the rationale to focus on epithelial cells vs. fibroblasts in lamina propria is well-justified (line 80).

The need for the present study, and the hypothesis, are well-established with reference to existing literature, and potential research applications (e.g. availability of a transcriptome dataset) are strong. The clinical significance of this work is overstated in the 454-5.

The manuscript is well-organized and written with clarity.

The manuscript could be strengthened by considering the following:

1. Dehydration can likely impact mucus composition and biomechanics. If dehydration alters composition of mucus overlying the vocal folds, then could the observed molecular changes have resulted from changes in vocal fold surface, not systemic, hydration? This should be addressed given that systemic, not surface, hydration is the focus of the study (e.g. as noted in title).

In a related minor point: mucus is arguably the first barrier to protect the vocal folds, not epithelium (line 49)

2. It is often acknowledged that the relationship between systematic hydration and vocal fold hydration is unclear, despite the clinical and research observations that dehydrating conditions have negative impacts on vocal function. With respect to the methodology used in the study, what support is there for the assumption that rabbit vocal fold hydration levels were impacted by rapid reduction in weight loss (e.g. histology)? Similarly, is there biomechanical/ physiological evidence to suggest that vocal fold systemic hydration can decrease in 3-6 hours?

3. Consider justifying sample size.

4. Consider providing a rationale for inducing mild dehydration (as indicated by a 5% loss of body weight) with specific respect to vocal fold biology, the focus of this study. The two references do not justify mild dehydration with respect to vocal folds (22,66).

Reviewer #2: This is a very interesting paper on the effects of systemic dehydration on the genetic profile of the vocal folds. This information is potentially useful to better understand the effects of systemic dehydration on voice production. The findings that cell adhesion genes are downregulated and cell reproduction upregilated are new and interesting, with a very thorough discussion. The paper adds valuable comments and balanced observations and conclusions in a well written discussion section. erhaps more information about the animals would be useful, such as their weight and morphometric measures if they are available. Rabbits are variable and thre may be confounding anatomical factors. It could also be useful to add specific examples of how the information could be useful in voice therapy.

Reviewer #3: This is an interesting study and a much-needed contribution to the field. The RNA-seq dataset could become an important resource to other researchers in epithelial biology and vocal fold biology. Overall the manuscript is very well-written and figures are clear. I have the following suggestions and concerns.

Major concerns

-As written, the experiment did not test the role of each vocal fold layer as described in the introduction, lines 84-85. This phrasing led me to expect separate results for epithelium, lamina propria, and maybe muscle. However, results are not parsed by tissue layer: only epithelium is extensively discussed, lamina propria is mentioned briefly in the discussion, and muscle is not mentioned.

-Please describe laryngeal and vocal fold dissection, specifically which tissue layers and extent of the length of the vocal folds were harvested for analysis. Without knowing details of the dissection it’s difficult to interpret whether DE genes were primarily involved in epithelial processes, or if predominance of epithelial processes in functional terms and genes selected for further analysis was an authorial decision.

-Line 404-405: If all that is known about NR4A3 in vocal folds is in fibroblasts, some more information about other upregulated epithelial cell proliferation markers post dehydration would better support claims regarding a new role for a family of transcription factors.

-Could furosemide itself have side effects on vocal folds? Please discuss feasibility of acute and/or chronic dehydration due to water withholding alone in the rabbit model.

Minor concerns

Abstract

-Slightly long. The first few sentences could be cut or edited down.

Introduction

-Line 63: Unclear phrasing; add “increased” before “physical activity.”

-Lines 94-95: This is a very broad hypothesis. I understand that RNA-seq is a hypothesis-generating methodology, but were there hypotheses for specific changes in different tissue layers?

Methods

-Please explain rationale for saline injection in control animals. To control for stress of handling and IP injections? Could the saline have changed hydration? For related reasons, please describe volumes of furosemide and saline injectates.

Results

-Line 224-225: It’s not clear that urine and stool were the only mechanisms of weight loss in the control group. Handling stress mentioned above could have decreased water and food consumption.

-Figure 1A: Please disclose exact p-value in caption or body text.

-How many animals received more than one injection of furosemide? Were there any differences in blood analytes, or would differences be expected?

-Figure 4 is minimally described. Some interpretation in this section regarding genes of interest in the other experiments would be helpful.

Discussion

-Line 353: Citation for the characterization of 5% acute dehydration as “mild”?

-Are there any other extant RNA-seq data on other mucosal tissues after dehydration? Please discuss.

-Paragraph from lines 353-381: Phonotrauma, LPR, and dehydration exist at varying levels of chronicity. Are all of the studies discussed comparable to the acute challenge in the present experiment?

Conclusions

-The conclusion (and introduction) mentions the ability to personalize hydration recommendations for vocal fold health, but these results do not yet support that. Hypotheses have been generated re: interactions of systemic dehydration and phonotrauma, LPR, wound healing, and infection, but clinical applicability is still limited.

6. PLOS authors have the option to publish the peer review history of their article (what does this mean?). If published, this will include your full peer review and any attached files.

Reviewer #1: No

Reviewer #2: Yes: Luc Mongeau

Reviewer #3: No

---

## [Author Response · Author response to Decision Letter 0]

16 Jun 2020

Dear Dr. Jetté,

 We thank the Editor and reviewers for the favorable comments on the submitted manuscript. We greatly appreciate the comprehensive and insightful questions of the editor and reviewers and the opportunity to improve the quality of our original submission. Please find the reviewers’ comments/questions and our responses below. The line numbers correspond to the ‘Revised Manuscript with Track Changes’.

PLOS ONE

Journal Requirements:

The PLOS ONE style templates can be found at

Response: All files were verified to meet PLOS ONE style requirements.

2. In your Methods, please provide scientific justification for the individual housing of the rabbit.

Response: We added a justification and reference for individual housing of male rabbits, and social interaction with humans as part of an enrichment protocol (lines 129-134).

“Individual housing is recommended for male rabbits, which should be separated from other males at sexual maturity (12 to 14 weeks) to avoid aggressive behavior, but maintaining visual and olfactory contact with other rabbits (https://www.nc3rs.org.uk/3rs-resources/housing-and-husbandry/rabbits) (Refinements in rabbit husbandry. Lab Anim. 27: 301-329.)”. Socialization training, consisting of food enrichment (hay) and human interaction (petting) was part of the acclimatization protocol to reduce stress for the rabbits and facilitate animal handling at the time of the study.”

3. We note that you are reporting an analysis of a microarray, next-generation sequencing, or deep sequencing data set. PLOS requires that authors comply with field-specific standards for preparation, recording, and deposition of data in repositories appropriate to their field. Please upload these data to a stable, public repository (such as ArrayExpress, Gene Expression Omnibus (GEO), DNA Data Bank of Japan (DDBJ), NCBI GenBank, NCBI Sequence Read Archive, or EMBL Nucleotide Sequence Database (ENA)). In your revised cover letter, please provide the relevant accession numbers that may be used to access these data. For a full list of recommended repositories, see http://journals.plos.org/plosone/s/data-availability#locomics or http://journals.plos.org/plosone/s/data-availability#loc-sequencing

Response: RNA-Seq data were deposited in the Gene Expression Omnibus (GEO) repository under the accession number GSE132765.

To review GEO accession GSE132765:

Go to https://www.ncbi.nlm.nih.gov/geo/query/acc.cgi?acc=GSE132765

Enter token arotaeqqdzwxruj into the box.

Review Comments to the Author

Reviewer #1: The main claims of this manuscript are that rapidly induced mild dehydration of rabbits induce molecular changes in vocal fold epithelium, and these changes are consistent with decreased homeostasis. Thus, the study seeks to elucidate a mechanism by which hydration can impact vocal fold integrity and, consequently, function. This work is significant in that it takes an incremental step towards identifying the molecular mechanism underlying well-established deleterious impacts of dehydration on vocal health (e.g., as stated on line 72). While vocal fold health depends on integrity of the epithelium, and underlying layers, the rationale to focus on epithelial cells vs. fibroblasts in lamina propria is well-justified (line 80). The need for the present study, and the hypothesis, are well-established with reference to existing literature, and potential research applications (e.g. availability of a transcriptome dataset) are strong. The clinical significance of this work is overstated in the 454-5. The manuscript is well-organized and written with clarity.

The manuscript could be strengthened by considering the following:

1. Dehydration can likely impact mucus composition and biomechanics. If dehydration alters composition of mucus overlying the vocal folds, then could the observed molecular changes have resulted from changes in vocal fold surface, not systemic, hydration? This should be addressed given that systemic, not surface, hydration is the focus of the study (e.g. as noted in title).

In a related minor point: mucus is arguably the first barrier to protect the vocal folds, not epithelium (line 49)

Response: Thank you very much for your comments and questions. To prevent/minimize surface dehydration from interfering with results, all rabbits in both control and systemically dehydrated groups were accommodated in the same controlled environment with optimal ambient temperature (15-20°C) and relative air humidity (~40%) recommended for rabbits (Guide for the Care and Use of Laboratory Animals. 8th edition. Washington (DC): National Academies Press (US); 2011). In this study, we did not analyze the mucus content on the surface of the vocal folds, but we did not find strong evidence on mucin changes based on gene expression. However, we cannot completely rule out that surface dehydration played a role in our findings. Our next study will focus on the proteomic analysis of the vocal folds using the same systemic dehydration rabbit model. Our group is currently investigating the effects of surface dehydration alone in the vocal folds using rabbit as a model. These two studies will allow us to have a better understanding of how these distinct types of dehydration intertwine at the molecular level. 

The authors agree that mucus, together with the epithelia, is the first barrier of protection of the vocal folds. We changed the text and added a reference in order to provide complete information to the readers (lines 60-61).

2. It is often acknowledged that the relationship between systematic hydration and vocal fold hydration is unclear, despite the clinical and research observations that dehydrating conditions have negative impacts on vocal function. With respect to the methodology used in the study, what support is there for the assumption that rabbit vocal fold hydration levels were impacted by rapid reduction in weight loss (e.g. histology)? Similarly, is there biomechanical/ physiological evidence to suggest that vocal fold systemic hydration can decrease in 3-6 hours?

Response: The literature reports systemic dehydration as low as 1% having a negative impact on vocal function. Based on these reports, the authors were mostly intrigued by how these negative clinical impacts would reflect in the vocal folds gene expression in response to a mild-moderate level of systemic dehydration. We believe that the difference in gene expression (203 genes) between the dehydrated and control groups support the hypothesis that the systemic dehydration, verified by body weight loss and hematologic changes, impacted the vocal folds in our model. In a previous study by our group, >6% body weight loss systemic dehydration by water withholding lead to vocal folds dehydration in an in vivo rat model (In Vivo Magnetic Resonance Imaging of the Rat Vocal Folds After Systemic Dehydration and Rehydration. J Speech Lang Hear Res. doi: 10.1044/2019_JSLHR-19-00062.). We understand the limitations of comparing both studies since they use different animal models and systemic dehydration methods. Still, these are relevant observations to address the questions raised by this reviewer. 

We did not apply histologic examination in this study. We do not expect to see histological changes in the rabbit vocal fold tissue at this relatively low dehydration level (5%) and short exposure (3-6h) based on previous laboratory observation of pilot studies. No evidence of inflammation or damage to the laryngeal mucosa was observed grossly nor by examination under the dissecting microscope by a certified veterinary pathologist. For this study, we did not evaluate biomechanical/ physiological features of the dehydrated vocal folds. Instead, we wanted to investigate how the changes in voice function observed in the literature could be explained at the molecular level. We believe that our results are a first step to give direction to further explore the effects of systemic dehydration on vocal fold biology.

3. Consider justifying sample size.

Response: The sample size was determined upon consultation with Purdue Bioinformatics Core in order to have the minimum ideal sample size required for the RNA-Seq statistical analysis. Our goal was also to limit the group sizes to the minimum needed to obtain statistical significance complying with the concept of reduction in animal research (“Three R’s - Replacement, Reduction, and Refinement.”).

4. Consider providing a rationale for inducing mild dehydration (as indicated by a 5% loss of body weight) with specific respect to vocal fold biology, the focus of this study. The two references do not justify mild dehydration with respect to vocal folds (22,66).

Response: The 5% systemic dehydration (based on body weight loss) was chosen as an altered hydration state physiologically translatable to the level of fluid loss associated with increased physical activities that humans may experience (Dehydration: physiology, assessment, and performance effects. Compr Physiol. doi:10.1002/cphy.c130017). As systemic dehydration can potentially have effects in various body functions, such as skin health, neurological function, gastrointestinal and renal functions (Narrative Review of Hydration and Selected Health Outcomes in the General Population. Nutrients. doi:10.3390/nu11010070), our central question was whether a 5% systemic dehydration would impact the vocal fold tissue at the molecular level in our animal model. 

A reference for levels of systemic dehydration related to vocal fold tissue was added: Oleson S, Cox A, Liu Z, Sivasankar MP, Lu KH. In Vivo Magnetic Resonance Imaging of the Rat Vocal Folds After Systemic Dehydration and Rehydration. J Speech Lang Hear Res. 2020;63(1):135‐142. doi:10.1044/2019_JSLHR-19-00062 (line 260).

As recommended by the reviewer, we have tempered the language of the clinical significance statement of the study in the Abstract (lines 41-43) and Conclusions (lines 518-520).

Reviewer #2: This is a very interesting paper on the effects of systemic dehydration on the genetic profile of the vocal folds. This information is potentially useful to better understand the effects of systemic dehydration on voice production. The findings that cell adhesion genes are downregulated and cell reproduction upregulated are new and interesting, with a very thorough discussion. The paper adds valuable comments and balanced observations and conclusions in a well written discussion section. Perhaps more information about the animals would be useful, such as their weight and morphometric measures if they are available. Rabbits are variable and there may be confounding anatomical factors. It could also be useful to add specific examples of how the information could be useful in voice therapy. 

Response: Thank you very much for your comments and recommendations. In order to minimize the difference between rabbits, they were matched by breed, sex and age. We added the information about the body weight (average and range) of all rabbits in the Materials and methods section (lines 139-140).

We have tempered the language of the clinical significance in the Conclusions to clarify that although our findings represent evidence to support the inclusion of hydration therapy as a clinical practice to maintain vocal health, it is still early to recommend specific therapies based on our results alone (lines 518-520).

Reviewer #3: This is an interesting study and a much-needed contribution to the field. The RNA-seq dataset could become an important resource to other researchers in epithelial biology and vocal fold biology. Overall the manuscript is very well-written and figures are clear. I have the following suggestions and concerns. 

Major concerns

-As written, the experiment did not test the role of each vocal fold layer as described in the introduction, lines 84-85. This phrasing led me to expect separate results for epithelium, lamina propria, and maybe muscle. However, results are not parsed by tissue layer: only epithelium is extensively discussed, lamina propria is mentioned briefly in the discussion, and muscle is not mentioned.

Response: Thank you for the opportunity to clarify this concern. The initial transcriptome analysis resulted in a total of 23,669 genes that were mapped to the rabbit’s genome, and includes genes expressed in muscle, lamina propria, and epithelia. The vast majority of these genes showed no difference in expression between control and systemically dehydrated groups, except for the 203 genes reported in the study. Interestingly, a subset of these differentially expressed genes was represented by epithelial related genes, which directed our focus to the epithelia. We added this information to the Results section to justify our focus on the epithelial processes (lines 290-292, and 302-303).

-Please describe laryngeal and vocal fold dissection, specifically which tissue layers and extent of the length of the vocal folds were harvested for analysis. Without knowing details of the dissection it’s difficult to interpret whether DE genes were primarily involved in epithelial processes, or if predominance of epithelial processes in functional terms and genes selected for further analysis was an authorial decision.

Response: Thank you for highlighting the lack of clarity on the dissection procedure. We added a paragraph explaining the dissection of the larynx and collection of the vocal fold tissue in the Materials and methods (lines 169-181). We collected the full extension of the vocal fold tissue with all layers (epithelia, LP, and muscle) represented in the excised section.

-Line 404-405: If all that is known about NR4A3 in vocal folds is in fibroblasts, some more information about other upregulated epithelial cell proliferation markers post dehydration would better support claims regarding a new role for a family of transcription factors. 

Response: Thank you for highlighting this. We modified the body text to clarify that more investigation is needed to understand the role of NR4A3 in the vocal fold epithelium as it relates to systemic dehydration (lines 454-455). Based on our findings, only one more gene involved in epithelial cell proliferation was upregulated in the RNA-Seq analysis.

-Could furosemide itself have side effects on vocal folds? Please discuss feasibility of acute and/or chronic dehydration due to water withholding alone in the rabbit model.

Response: The effects of furosemide are characterized in tissues such as kidney, adrenal glands, liver, lungs, and spleen (Pharmacokinetic, biliary excretion, and metabolic studies of 14C-furosemide in the rat. Xenobiotica. doi:10.3109/00498259109039512). In addition, anti-inflammatory effects have been reported in human mononuclear cells (Immunosuppressive and cytotoxic effects of furosemide on human peripheral blood mononuclear cells. Ann Allergy Asthma Immunol. doi:10.1016/S1081-1206(10)62870-0), and lungs when the respiratory route of administration is used (Furosemide: progress in understanding its diuretic, anti-inflammatory, and bronchodilating mechanism of action, and use in the treatment of respiratory tract diseases. Am J Ther. doi:10.1097/00045391-200207000-00009). To the best of our knowledge there is no literature linking furosemide effects directly to vocal folds; however, a possible effect cannot be completely ruled out. Nevertheless, our study shows that the two genes encoding the Na+2Cl-K+ cotransporter (SLC12A1 and SLC12A2), which is the receptor for furosemide, are present in the vocal folds of both euhydrated and dehydrated groups with no differential expression. In addition, other sodium channels and genes reported to be possibly affected by furosemide such as IL6, IL8, TNFα, prostaglandins, leukotrienes, are also present in both transcriptomes with no differential expression. Collectively, these findings support that there is no evidence for a direct or indirect effect of furosemide alone in the vocal folds in our rabbit model. 

It is feasible to perform experiments based on water withholding, and animal models using this method are well established, especially for the study of longer periods of dehydration (references listed below). However, based on our experience, the time needed for the rabbit to achieve the desired 5% dehydration with water withholding would be at least 24h, which would not represent an acute situation simulating an increased physical activity for example. Similar results are seen in the rat model; one of our studies in rats showed that the time range to achieve mild systemic dehydration after water withholding was 18–24h, for moderate dehydration was 36–48h, and for marked dehydration was 66–72h (In Vivo Magnetic Resonance Imaging of the Rat Vocal Folds After Systemic Dehydration and Rehydration. J Speech Lang Hear Res. doi:10.1044/2019_JSLHR-19-00062). Thus, the decision to use furosemide was to guarantee we were inducing an acute state of dehydration. In addition, the decreased experimental time contributes to the animals’ welfare. Interestingly, our group observed a similar pattern of gene expression changes in the vocal fold in a rat model of systemic dehydration by water restriction (Restricted Water Intake Adversely Affects Rat Vocal Fold Biology, accepted on May 27/2020, The Laryngoscope). In that study, the gene expression and protein levels of desmoglein 1, and the gene expression of IL1A were also downregulated in the dehydrated group, which supports the effect of systemic dehydration on the expression of these genes despite the dehydration protocol used.

Water withholding studies references:

- Lee MH, Choi HY, Sung YA, Lee JK. High signal intensity of the posterior pituitary gland on T1-weighted MR images. Correlation with plasma vasopressin concentration to water deprivation. Acta Radiol. 2001;42(2):129‐134. doi:10.1034/j.1600-0455.2001.042002129.x

- Islam S, Abély M, Alam NH, Dossou F, Chowdhury AK, Desjeux JF. Water and electrolyte salvage in an animal model of dehydration and malnutrition. J Pediatr Gastroenterol Nutr. 2004;38(1):27‐33. doi:10.1097/00005176-200401000-00009

- Kishore BK, Krane CM, Miller RL, et al. P2Y2 receptor mRNA and protein expression is altered in inner medullas of hydrated and dehydrated rats: relevance to AVP-independent regulation of IMCD function. Am J Physiol Renal Physiol. 2005;288(6):F1164‐F1172.

- Cox A, Cannes do Nascimento N, Pires Dos Santos A, Sivasankar MP. Dehydration and Estrous Staging in the Rat Larynx: an in vivo Prospective Investigation [published online ahead of print, 2019 Jul 13]. J Voice. 2019;S0892-1997(19)30236-X. doi:10.1016/j.jvoice.2019.06.009

- Oleson S, Cox A, Liu Z, Sivasankar MP, Lu KH. In Vivo Magnetic Resonance Imaging of the Rat Vocal Folds After Systemic Dehydration and Rehydration. J Speech Lang Hear Res. 2020;63(1):135‐142. Published 2020 Jan 10. doi:10.1044/2019_JSLHR-19-00062

Minor concerns

Abstract

-Slightly long. The first few sentences could be cut or edited down. 

Response: We edited down the first sentences (six to three lines) of the abstract to make it more concise (lines 25-27).

Introduction

-Line 63: Unclear phrasing; add “increased” before “physical activity.”

Response: Corrected (line 74).

-Lines 94-95: This is a very broad hypothesis. I understand that RNA-seq is a hypothesis-generating

methodology, but were there hypotheses for specific changes in different tissue layers?

Response: We hypothesized that changes in gene expression would occur in different vocal fold tissue layers based on physiological and biomechanical changes in vocal folds due to systemic dehydration and other challenges reported in the literature. Some candidate gene categories that we expected to see changes included but were not limited to: components of extracellular matrix involved in lubrication and viscoelasticity of the vocal fold such as hyaluronan and collagen; epithelial related genes such as components of cell junctions (cadherin and claudins), ion transporters, mucins and aquaporins; and muscle related genes such as myosin heavy chain genes, which are a major determinant of skeletal muscle physiology. We modified the text in the Introduction to make the hypothesis less broad, but without restricting to specific genes (lines 105-107).

Methods

-Please explain rationale for saline injection in control animals. To control for stress of handling and IP injections? Could the saline have changed hydration? For related reasons, please describe volumes of furosemide and saline injectates.

Response: The saline injection in control rabbits was applied to prevent animal handling and IP injections from being a confounding variable and an unwanted source of variation between groups (https://www.nc3rs.org.uk/handling-and-restraint). Saline was chosen as a physiological solution, and also for being in the composition of Salix solution (commercial furosemide brand) used to stimulate systemic dehydration in this study. The concentration of the commercial furosemide solution is 50 mg/mL, to achieve a 5% dehydration the volume of furosemide was calculated based on the body weight of each rabbit; an average volume of 0.34 mL was IP injected in each rabbit. The same calculation was used for saline. Based on this small volume of injectates and the results of our study (no significant differences in body weight and hematologic values in controls), we do not think that saline changed the hydration status of control rabbits. The rationale for saline injection and volume of injectates was added to the body text (lines 142-146). 

Results

-Line 224-225: It’s not clear that urine and stool were the only mechanisms of weight loss in the control group. Handling stress mentioned above could have decreased water and food consumption.

Response: We agree with the reviewer that handling stress could have affected the overall behavior of the control rabbits regarding food and water intake. We added this information in the text (lines 261-263). We would like to point out that the experiments were performed during the day and rabbits are nocturnal animals, having higher activity during nighttime (Biology of the rabbit. J Am Assoc Lab Anim Sci. 2006;45(1):8‐24.). Moreover, no hematological changes (PCV, TPP, BUN, creatinine, etc.) were observed in the control rabbits when comparing pre-IP saline injection and pre-euthanasia values, indicating that systemic dehydration was not induced in this group despite the slight body weight loss observed.

-Figure 1A: Please disclose exact p-value in caption or body text.

Response: Exact p-value (p= 0.0003) of comparison between groups in Fig 1A (body weight loss) was added to the body text (line 263). 

-How many animals received more than one injection of furosemide? Were there any differences in blood analytes, or would differences be expected?

Response: In the control group, three rabbits received 1 saline injection and three received 2. In the dehydrated group, five rabbits received 1 furosemide injection and three received 2. The number of injections was matched between control and dehydrated rabbits assigned to the experiment in the same day (with 2-4 rabbits/experiment/day). We would expect sodium and chloride blood levels to be lower in rabbits that received two injections of furosemide due to the mechanism of action of this diuretic, which inhibits Na+ and Cl- reabsorption. However, no differences were observed, possibly because rabbits were euthanized when they reached ~5% dehydration regardless of receiving 1 or 2 injections. No differences were observed within the control group either. We attribute the faster or slower diuretic response to furosemide to individual metabolism of each rabbit. 

-Figure 4 is minimally described. Some interpretation in this section regarding genes of interest in the other experiments would be helpful.

Response: We focused the discussion on the clusters with a higher number of predicted protein interactions, and with a putative role in the epithelial structure maintenance. We expanded the discussion to include more of these proteins (lines 483-486, and 495-502). We hope this network can be used to explore other mechanisms involved in the response of vocal folds to systemic dehydration.

Discussion

-Line 353: Citation for the characterization of 5% acute dehydration as “mild”?

Response: Citation for 5% dehydration as mild was added: Oleson S et al. In Vivo Magnetic Resonance Imaging of the Rat Vocal Folds After Systemic Dehydration and Rehydration. J Speech Lang Hear Res. 2020;63(1):135‐142. (line 399).

-Are there any other extant RNA-seq data on other mucosal tissues after dehydration? Please discuss.

Response: To the best of our knowledge, studies on RNA-Seq transcriptome of mucosal tissue exposed to dehydration are not available in the literature. Other tissues have been explored regarding their transcriptional changes due to dehydration, such as kidney, brain (subfornical organ, and supraoptic nucleus of the hypothalamus), and testes. Transcriptome studies on airway and intestinal mucosa in conditions where dehydration of the tissue surface is associated with defects in mucus clearance, which is characteristic of mucoobstructive pulmonary diseases (e.g., cystic fibrosis, primary ciliary dyskinesia, and the chronic bronchitic form of chronic obstructive pulmonary disease) are reported in the literature, but those are not comparable to our study. 

-Paragraph from lines 353-381: Phonotrauma, LPR, and dehydration exist at varying levels of chronicity. Are all of the studies discussed comparable to the acute challenge in the present experiment?

Response: The studies regarding phonotrauma discussed in the present study reflect data collected at various time points using different animal models subjected to 30 minutes (rabbits), 2-4 hours (dogs), or 25 minutes during 2-15 weeks (cats) of increased phonation. These studies are comparable to our acute systemic dehydration study. The LPR study with human subjects presents data from vocal fold biopsies of patients previously diagnosed with LPR and, therefore, would be considered chronic when compared to our study. We clarified this in the text (lines 423-425).

Conclusions

-The conclusion (and introduction) mentions the ability to personalize hydration recommendations for vocal fold health, but these results do not yet support that. Hypotheses have been generated re: interactions of systemic dehydration and phonotrauma, LPR, wound healing, and infection, but clinical applicability is still limited.

Response: Thank you for your comment. We acknowledge the limitations of our results regarding the clinical applicability in voice therapy. We recognize that the transcriptome data is just a start to improve the understanding of how hydration impacts the vocal fold biology, and additional studies addressing the impact of systemic dehydration associated with other conditions such as phonotrauma, LPR, wound healing, etc. are warranted. Therefore, we modified this conclusion statement in the Abstract (lines 41-43) and Conclusions (lines 518-520).

---

## [Decision Letter · Decision Letter 1]

7 Jul 2020

Unraveling the molecular pathobiology of vocal fold systemic dehydration using an in vivo rabbit model

PONE-D-20-11397R1

Dear Dr. Cannes do Nascimento,

We’re pleased to inform you that your manuscript has been judged scientifically suitable for publication and will be formally accepted for publication once it meets all outstanding technical requirements.

Kind regards,

Marie Jetté

Academic Editor

PLOS ONE

Additional Editor Comments (optional):

Reviewers' comments:

Reviewer's Responses to Questions

**Comments to the Author**

1. If the authors have adequately addressed your comments raised in a previous round of review and you feel that this manuscript is now acceptable for publication, you may indicate that here to bypass the “Comments to the Author” section, enter your conflict of interest statement in the “Confidential to Editor” section, and submit your "Accept" recommendation.

Reviewer #1: All comments have been addressed

Reviewer #2: All comments have been addressed

Reviewer #3: All comments have been addressed

2. Is the manuscript technically sound, and do the data support the conclusions?

Reviewer #1: Yes

Reviewer #2: Yes

Reviewer #3: Yes

3. Has the statistical analysis been performed appropriately and rigorously? 

Reviewer #1: Yes

Reviewer #2: Yes

Reviewer #3: Yes

4. Have the authors made all data underlying the findings in their manuscript fully available?

Reviewer #1: Yes

Reviewer #2: Yes

Reviewer #3: Yes

5. Is the manuscript presented in an intelligible fashion and written in standard English?

Reviewer #1: Yes

Reviewer #2: Yes

Reviewer #3: Yes

6. Review Comments to the Author

Reviewer #1: This revised manuscript, describing original research, was strengthened by the authors' thorough responses to concerns. Methods are well-described, and justified in the reviewer responses. The revised conclusions are supported by data. I have no additional concerns.

Reviewer #2: (No Response)

Reviewer #3: This paper is substantially improved. Methodology is much more completely described, rationale is clearer, and conclusions are supported by findings. All of my concerns have been addressed. I question whether the very nice explanation in the authors’ response to reviewers regarding lack of evidence for effects of furosemide in vocal folds should be included in the paper somewhere. In my opinion, this would strengthen the paper because readers may have the same question as I did, but is not strictly necessary.

7. PLOS authors have the option to publish the peer review history of their article (what does this mean?). If published, this will include your full peer review and any attached files.

Reviewer #1: No

Reviewer #2: No

Reviewer #3: No

---

## [Editor Report · Acceptance letter]

22 Jul 2020

PONE-D-20-11397R1 

Unraveling the molecular pathobiology of vocal fold systemic dehydration using an in vivo rabbit model 

Dear Dr. Cannes do Nascimento:

I'm pleased to inform you that your manuscript has been deemed suitable for publication in PLOS ONE. Congratulations! Your manuscript is now with our production department. 

Kind regards, 

on behalf of

Dr. Marie Jetté 

Academic Editor

PLOS ONE